# Pollutant Removal Efficiency of a Bioretention Cell with Enhanced Dephosphorization

**Chia-Chun Ho * and Yi-Xuan Lin**

Department of Civil and Construction Engineering, National Taiwan University of Science and Technology, Taipei 106, Taiwan; smile917006@gmail.com
* Correspondence: cchocv@mail.ntust.edu.tw; Tel.: +886-2-2730-1073

**Abstract:** Low impact development can contribute to Sustainable Development Goals (SDGs) 2, 6, 7, 11, and 13, and bioretention cells are commonly used to reduce nonpoint source pollution. However, although bioretention is effective in reducing ammonia nitrogen and chemical oxygen demand (COD) pollution, it performs poorly in phosphorus removal. In this study, a new type of enhanced dephosphorization bioretention cell (EBC) was developed; it removes nitrogen and COD efficiently but also provides excellent phosphorus removal performance. An EBC (length: 45 m; width: 15 m) and a traditional bioretention cell (TBC) of the same size were constructed in Anhui, China, to treat rural nonpoint source pollution with high phosphorus concentration levels. After almost 2 years of on-site operation, the ammonium nitrogen removal performance of the TBC was 81%, whereas that of the EBC was 78%. The COD removal rates of the TBC and EBC were 51% and 65%, and they removed 51% and 92% of the total phosphorus, respectively. These results indicate that the TBC and EBC have similar performance in the removal of ammonium nitrogen and COD, but the EBC significantly outperforms the TBC in terms of total phosphorus removed.

**Keywords:** low impact development; Sustainable Development Goals; non-point source pollution; enhanced dephosphorization bioretention

## 1. Introduction

Rapid urbanization is increasingly affected by extreme weather events. The frequent occurrence of short-duration intense rainfall creates pollution sources such as surface runoff, which lead to urban nonpoint source pollution [1]. China is affected not only by urban nonpoint source pollution but also by the environmental pollution caused by runoff rainwater in rural areas. Rural runoff rainwater contains nonpoint source pollution from human activities, livestock, and agriculture. Therefore, the pollution concentration in rural runoff is higher than that of urban runoff, particularly in terms of nutrients [2].

Low impact development (LID) practices can purify and reintegrate contaminated runoff into the hydrological cycle by increasing infiltration, reducing runoff velocity, and reducing pollutant load [3]. Macedo et al. [4] proposed that a new-generation LID model with the capacity to contribute to Sustainable Development Goals (SDGs) 2 (zero hunger), 6 (clean water and sanitation), 7 (affordable and clean energy), 11 (sustainable cities and communities), and 13 (climate action) [5]. The implementation of LID practices that align with SDGs is guided by the phrase "Think globally, act locally," which is increasingly used in the lexicon of sustainable development [6]. Bioretention cells are commonly used to reduce nonpoint source pollution because of their runoff and pollution control capabilities and their role as a landscape feature.

Bioretention was developed in Prince George County, Maryland, USA, in the 1990s [7]. Bioretention is an LID approach that can be used to address nonpoint source pollution [8]. The composition of a typical bioretention system includes plantings, a mulch layer, planting soil, filter fabric, a sand layer, a gravel bed, an outflow pipe, and geotextiles (Figure 1).

Bioretention mainly uses processes such as filtration, adsorption, plant uptake, and biological transformation to reduce the pollutants generated by runoff [9]. Most studies that aimed to improve the pollution removal ability of bioretention have focused on changing the composition of the planting soil (e.g., changing the ratio of perlite to vermiculite) [10]. Le Coustumer et al. [11] reported that adding compost to planting soil can increase the water permeability and pollutant removal rate of bioretention; the improvement in performance achieved through this method is greater than that achieved by adding 10% vermiculite and 10% perlite to planting soil. In general, the effect of bioretention on phosphorus reduction is relatively unstable, and it can easily cause clogging in the long term [12].

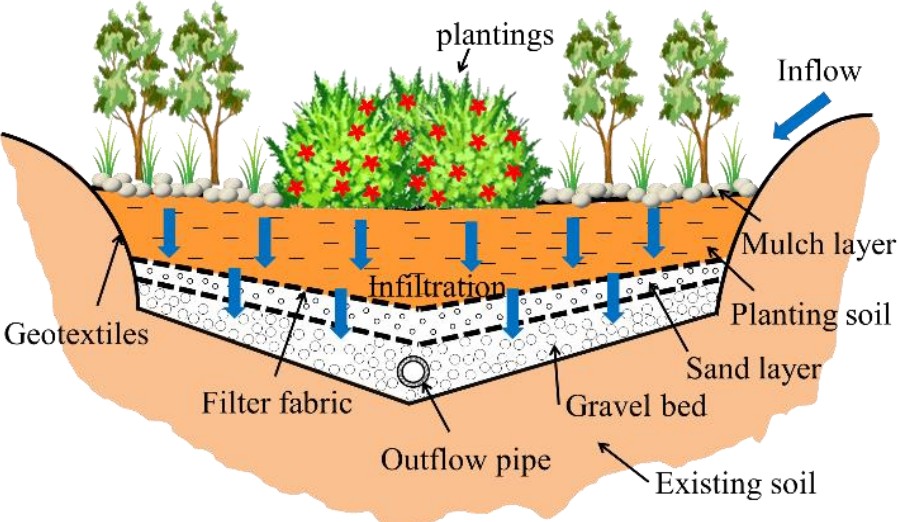

**Figure 1.** Typical bioretention system.

Fan et al. [13] proposed that if a proper submerged space is established inside a bioretention cell, the denitrification effect can be significantly increased. Li et al. [14] developed two groups of lab-scale bioretention cells. The first group of bioretention cells was filled with coarse gravel, medium gravel, fine gravel, and planting soil in order from bottom to top according to the model used for traditional bioretention cells (TBCs). For the second group of bioretention cells, the order of the fillers was reversed, such that the coarse gravel became the top layer (Figure 2). The test results indicated that the TBC was slightly more effective than the inverted cell in removing ammonium nitrogen ($NH_4^+$), with the removal percentages of the traditional cell being between 96.6% and 99.7% and those of the inverted cell being between 80.5% and 97.4%. However, the inverted bioretention cell had a significantly higher nitrate ($NO_3^-$-N) removal efficiency than the traditional cell. This is because the inverted bioretention cell can form an anoxic zone at its bottom and accelerate the denitrification reaction. Moreover, a series of bioretention box experiments involving various runoff inflow characteristics were performed by Davis et al. [15]. Their experimental results indicated that increasing or decreasing the hydrogen ion concentration (pH) from a neutral level leads to the release of phosphorus in the upper soil, resulting in the poor phosphorus removal performance of bioretention cells.

Numerous studies have demonstrated that bioretention cells provide favorable nitrogen removal performance but poor phosphorus removal performance. For algal growth, nitrogen is generally a limiting nutrient in coastal and oceanic waters, whereas phosphorus tends to be a limiting nutrient in freshwater systems [16]. If runoff rainwater with a high phosphorus concentration flows directly into rivers or lakes, it may cause water eutrophication. To date, no study has demonstrated that TBCs have sufficient phosphorus removal efficiency. Therefore, the present study modified the materials inside a bioretention cell and developed an enhanced dephosphorization bioretention cell (EBC).

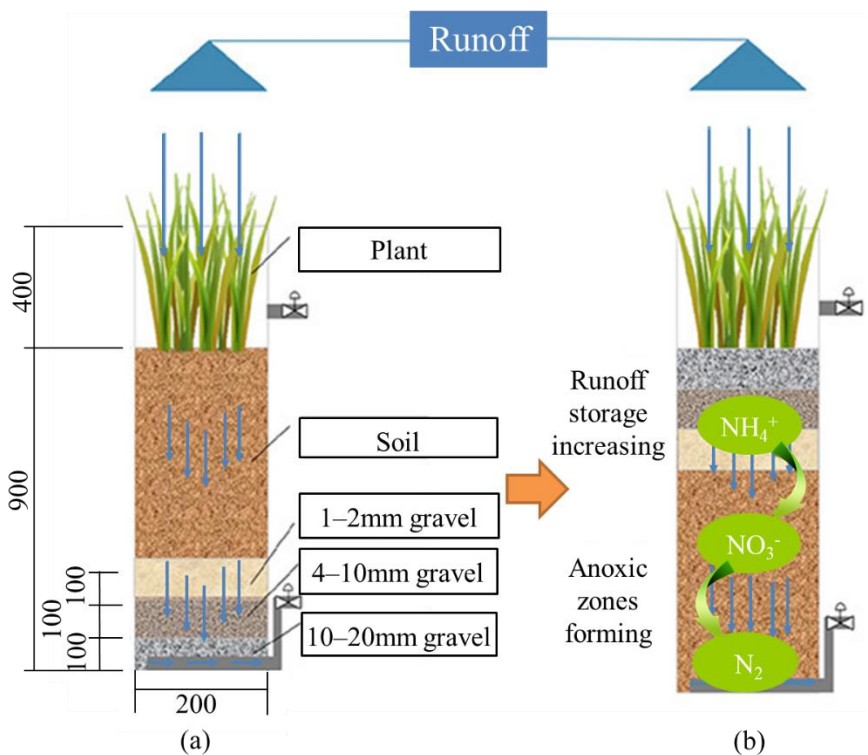

**Figure 2.** Profile of the retention cell (**a**) traditional, (**b**) inverted.

## 2. Materials and Methods

### 2.1. Enhanced Dephosphorization Bioretention Cell

The lowermost layer of the EBC is a mixed filter material layer instead of a gravel bed, as is the case in TBCs. The mixed filter material layer primarily comprises soil mixture layers (SMLs) and permeable layers (PLs). Figure 3 illustrates the enhanced dephosphorization bioretention system. The SMLs are 40 cm wide, 60 cm long, and 10 cm high, and they are stacked in layers. The SMLs are separated from each other by a 10 cm gap (at the top, bottom, left, and right sides of each SML) that is filled by the PL. The SMLs comprise approximately 70% to 80% of on-site soil mixed with approximately 20% to 30% of additional materials (e.g., active charcoal powder, organic matter, and iron).

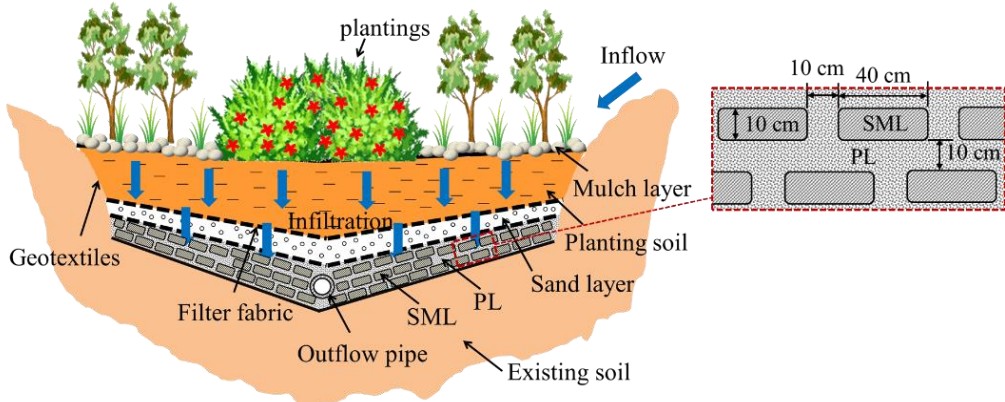

**Figure 3.** Enhanced dephosphorization bioretention cell (EBC). SML, soil mixture layer; PL, permeable layer.

Among the various materials that form SMLs, the soil serves as a habitat for microorganisms, and the active charcoal powder adsorbs high amounts of organic matter in wastewater, thus enhancing the efficiency of organic matter decomposition. The organic

matter (e.g., sawdust, straw, corn cobs, and kenaf) serves as nutrients for microorganisms, and the iron materials effectively adsorb phosphates. The materials are mixed and packed into fiber bags, which are then stacked to form the SMLs, with each layer being separated by a PL. A PL comprises aggregates of gravel, pumice, or zeolite measuring approximately 1–5 mm in diameter. Aggregates should be of consistent size to reduce the risk of clogging and to facilitate the dispersion of water in the system. Moreover, the surface of the aggregates that constitute a PL also serves as a habitat for nitrobacteria and adsorbs the organic matter in wastewater. Therefore, both layers actively remove pollutants from wastewater [17].

*2.2. Study Site*

Chaohu Lake, situated in the central region of Anhui Province, is the fifth largest shallow freshwater lake in China with a surface area of 775 km$^2$ and watershed area of 12,938 km$^2$. Chaohu Lake plays a key role in the social, economic, and ecological functions of the local basin. However, the excessive discharge of industrial and municipal wastewater caused by rapid industrialization and urbanization has led to the heavy pollution of Chaohu Lake [18–21]. To improve the water quality of Chaohu Lake, the nutrients flowing into the lake must first be reduced.

The study site is located to the northeast of Chaohu Lake (Figure 4). GPS coordinates of the study site are 31°40.01′ N, 117°40.73′ E. The sub-watershed area of the study site is approximately 520,000 m$^2$. It includes 440,000 m$^2$ of farmland and 80,000 m$^2$ of rural land. When it rains, the runoff flushes agricultural and rural nonpoint source pollution into the Jiyu River (which flows into Chaohu Lake), resulting in severe eutrophication in Chaohu Lake.

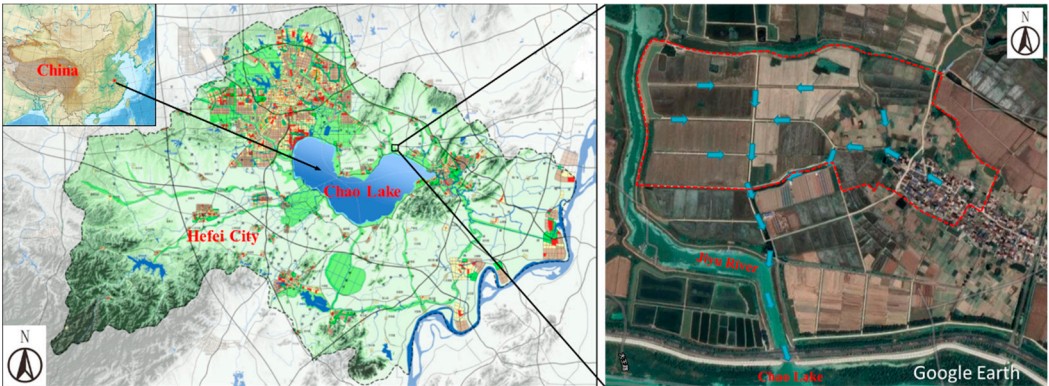

**Figure 4.** Location of study site in Chaohu Lake.

Agricultural nonpoint source pollution and urban domestic sewage are the main sources of nitrogen and phosphorus in Chaohu Lake [22,23]. In the experience of the authors, bioretention cells are often used to purify nonpoint source pollution. Therefore, in the present study, a TBC and an EBC were constructed on site, and their effectiveness in removing nutrients was compared. The two bioretention cells had the same dimensions (45 m length, 15 m width, and 1.2 m depth). Detailed cross-sectional views of the two cells are illustrated in Figure 5.

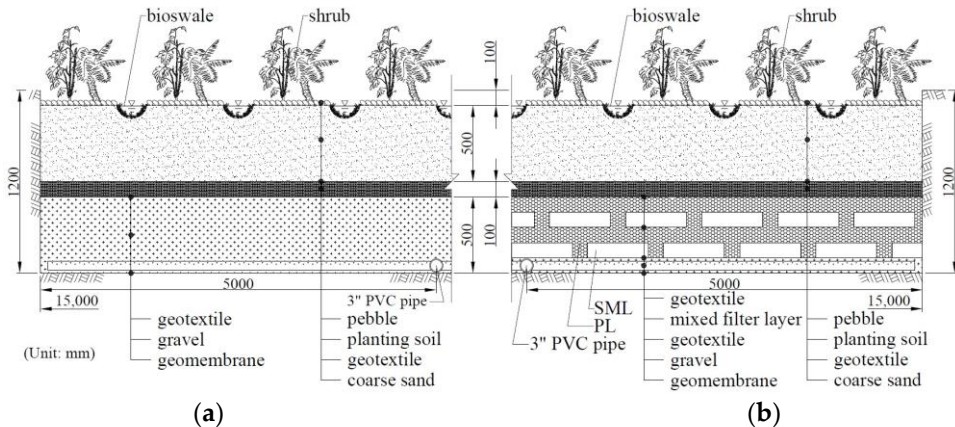

**Figure 5.** Cross-sectional view of bioretention cells: (**a**) traditional bioretention cell (TBC) and (**b**) EBC.

*2.3. Construction Steps*

A 675 m² TBC and a 675 m² EBC were constructed to treat the nonpoint source pollution and agricultural irrigation tailwater generated in an area measuring 520,000 m². The treatment rate of this case was 1:770, and the construction steps were as follows.

1. Excavation: Two adjacent pits with the same depth of 1.2 m were excavated at the site. Because the soil provided sufficient strength, the excavation could be performed vertically and no earth-retaining facility was required (see Figure 6a).
2. Installation of geomembrane: A 2-mm-thick geomembrane was spread around the pit of each cell to prevent groundwater from entering the cells (see Figure 6b).
3. Filling of cells with gravel: The TBC and EBC were, respectively, filled with 500-mm- and 100-mm-thick gravel layers (see Figure 6c).
4. Installation of outflow pipes: A polyvinyl chloride (PVC) pipe with a 3-in diameter was placed in the gravel layer and connected by a porous PVC pipe with a 2-in diameter (see Figure 6d).
5. Filling of SMLs and PLs: The inside of the EBC was filled with two layers of SMLs and two layers of PLs; the thickness of each layer was 100 mm (see Figure 6e).
6. Installation of geotextiles and filling of coarse sand: After geotextiles were laid on top of the gravel or PL, the cells were filled with a 100-mm-thick coarse sand layer (see Figure 6f).
7. Installation of geotextiles: Geotextiles were laid over the coarse sand layer to separate the planting soil and coarse sand layer (see Figure 6g).
8. Filling of planting soil: The two cells were filled with a 500-mm-thick layer of planting soil (see Figure 6h).
9. Planting: *Photinia serratifolia*, *Ficus microcarpa* cv. Golden Leaves, and *Ehretia microphylla* Lamk were planted in the two bioretention cells (see Figure 6i).
10. Installation of a weir: To allow the water in the channel to flow into the retention tank, a weir was installed to raise the water level (see Figure 6j).
11. Installation of inlet pipes: 4-in PVC pipes were used to divert water from the weir into the bioretention cells (see Figure 6k).
12. Operation: Contaminated water was drained into the bioswales of the bioretention cells to allow for infiltration and purification. The purified water was then discharged into the river using gravity (see Figure 6l).

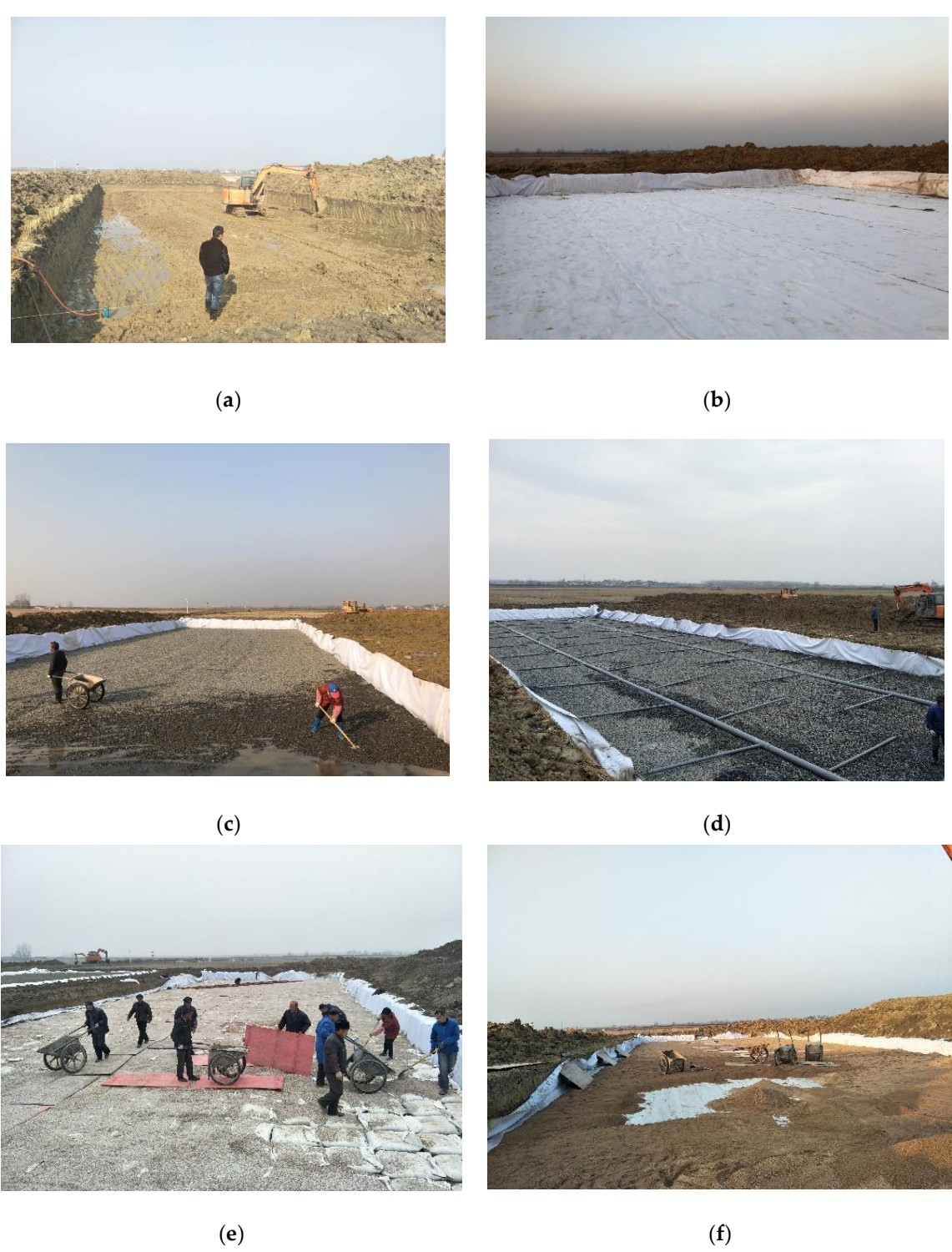

(**a**)　　　　　　　　　　　　　　　　　　　　(**b**)

(**c**)　　　　　　　　　　　　　　　　　　　　(**d**)

(**e**)　　　　　　　　　　　　　　　　　　　　(**f**)

**Figure 6.** *Cont.*

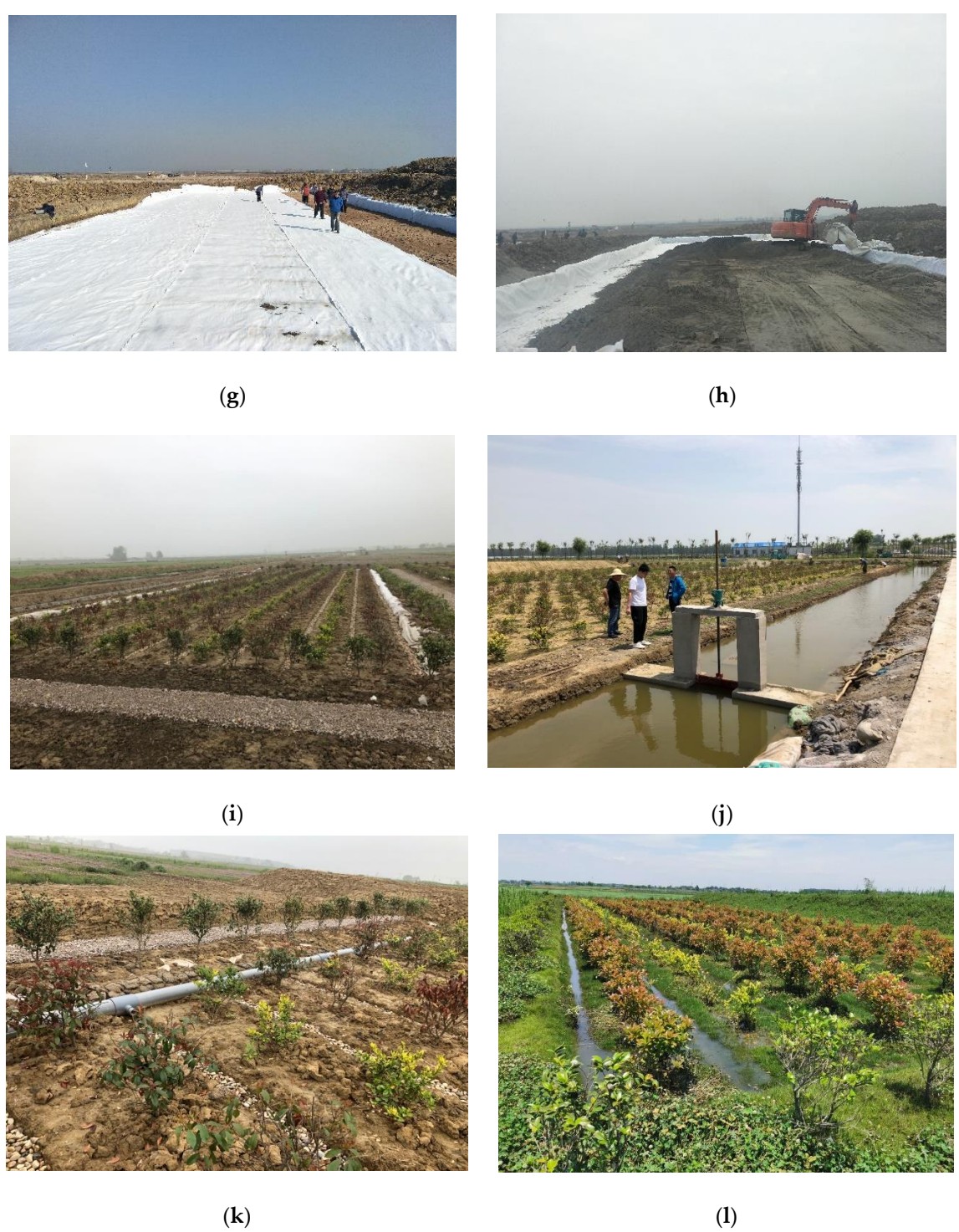

**Figure 6.** Construction steps: (**a**) excavation; (**b**) installation of geomembrane; (**c**) filling of gravel; (**d**) installation of outflow pipes; (**e**) filling of SMLs and PLs; (**f**) filling of coarse sand; (**g**) installation of geotextiles; (**h**) filling of planting soil; (**i**) planting; (**j**) installation of a weir; (**k**) installation of inlet pipes; (**l**) operation.

## 3. Results and Discussion

The two bioretention cells were completed in March 2019. To elucidate their effectiveness in removing pollutants, a 2-year sampling survey was conducted. Sampling was performed once a month, and a total of 24 samples were collected for analysis. Suspended

solids (SS), chemical oxygen demand (COD), ammonium nitrogen ($NH_4^+$-N), total nitrogen (TN), total phosphorus (TP), and phosphate ($PO_4$) were tested in the present study.

### 3.1. Suspended Solid

Table 1 and Figure 7 shows the test results for SS that were obtained through a 2-year experiment that was conducted between May 2019 and April 2021. The results indicate that the concentration of SS was higher during the rainy season from June to October. When contaminated water flows into the bioretention cells, SS decrease significantly. Because of the filtering function of the material inside the bioretention cells, they have a significant effect on SS removal. The average removal percentages of the TBC and EBC were 85% and 83%, respectively. The TBC and EBC have similar average removal percentages, indicating their favorable performance for SS reduction. However, the bioswales and pebbles on the surface of the bioretention cells required regular cleaning to avoid clogging due to the excessive accumulation of mud. Moreover, Student's t-test was used to was used to determine whether the performance of TBC and EBC were different. For SS removal, the *T*-value of TBC and EBC is 47.5%. If the significance level ($\alpha = 5\%$) was used, the test result showed that there was no difference in SS removal performance between TBC and EBC.

**Table 1.** The concentration and removal percentage of SS during the 2-year experiment.

| Date | Concentration (mg/L) | | | Removal Percentage (%) | |
|---|---|---|---|---|---|
| | $C_i$ [1] | $C_{o,TBC}$ [2] | $C_{o,EBC}$ [3] | TBC | EBC |
| May 2019 | 51 | 7 | 10 | 86 | 80 |
| June 2019 | 34 | 4 | 5 | 88 | 85 |
| July 2019 | 61 | 5 | 7 | 92 | 89 |
| August 2019 | 44 | 4 | 7 | 91 | 84 |
| September 2019 | 32 | 4 | 4 | 88 | 88 |
| October 2019 | 31 | 4 | 8 | 87 | 74 |
| November 2019 | 21 | 0 | 4 | 100 | 81 |
| December 2019 | 22 | 4 | 0 | 82 | 100 |
| January 2020 | 19 | 2 | 4 | 89 | 79 |
| February 2020 | 7 | 5 | 4 | 29 | 43 |
| March 2020 | 18 | 0 | 2 | 100 | 89 |
| April 2020 | 18 | 0 | 2 | 100 | 89 |
| May 2020 | 42 | 5 | 2 | 88 | 95 |
| June 2020 | 77 | 10 | 8 | 87 | 90 |
| July 2020 | 83 | 8 | 4 | 90 | 95 |
| August 2020 | 69 | 11 | 8 | 84 | 88 |
| September 2020 | 66 | 8 | 12 | 88 | 82 |
| October 2020 | 54 | 10 | 6 | 81 | 89 |
| November 2020 | 48 | 4 | 8 | 92 | 83 |
| December 2020 | 32 | 8 | 4 | 75 | 88 |
| January 2021 | 29 | 5 | 4 | 83 | 86 |
| February 2021 | 21 | 8 | 2 | 62 | 90 |
| March 2021 | 37 | 4 | 8 | 89 | 78 |
| April 2021 | 19 | 4 | 4 | 79 | 79 |
| Average | 39 | 5 | 5 | 85 | 83 |
| *T*-test value | | | | 47.5 | |

[1] Inflow concentration. [2] Outflow concentration of TBC. [3] Outflow concentration of EBC.

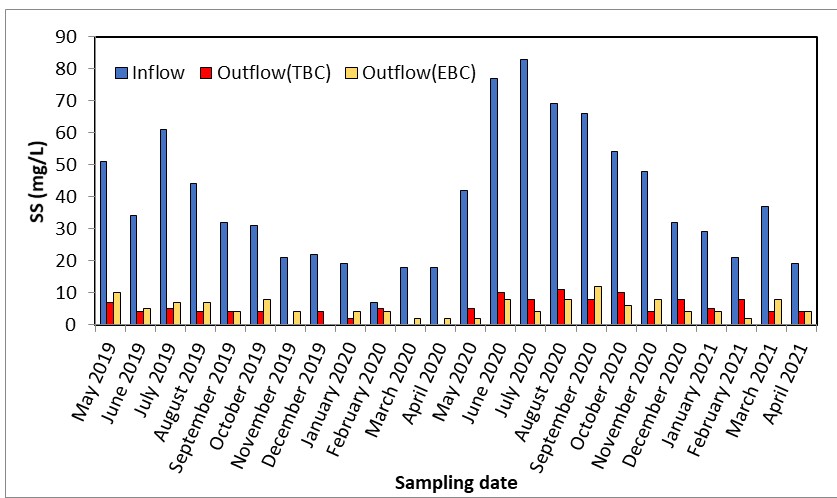

**Figure 7.** Suspended solid (SS) levels during the 2-year experiment.

*3.2. Chemical Oxygen Demand*

The test results for COD are presented in Table 2 and Figure 8. Both the TBC and EBC performed poorly for COD removal in the early stage of the facility's operation. This is because, at that stage, the microbial system within the bioretention cells was not fully established, and the ability of the microbes to biodegrade COD was not fully developed. However, after approximately half a year, stable pollution removal results were observed. The TBC's average pollution removal efficiency for COD was approximately 51%, and that of the EBC was 65%. The EBC outperformed the TBC, and the outflow concentration of the EBC was relatively stable. The first reason for this performance disparity is that the PLs in the EBC comprise zeolite, which can perform the function of adsorption. The second reason is that the SMLs in the EBC contains natural organic materials (e.g., rice stalks and rice husks), and the decomposed plant fiber can provide a carbon source and cultivate cellulose-degrading bacteria that biodegrades COD. However, both the TBC and EBC exhibited favorable COD biodegradation performance because the planting soil was sufficiently conducive to the cultivation of a microbial system. The *T*-value of COD removal between TBC and EBC was 0.2%. It showed that the performance of TBC and EBC in COD removal was different.

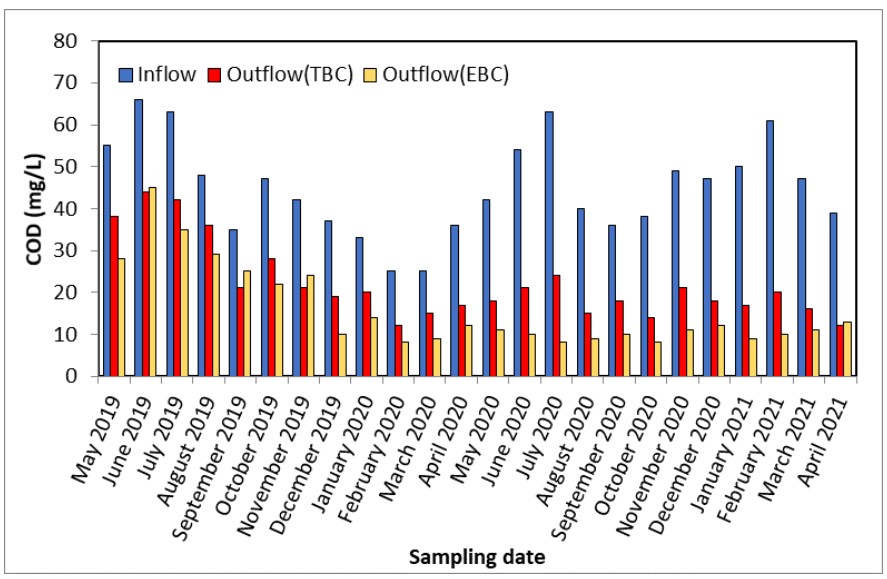

**Figure 8.** Chemical oxygen demand (COD) during the 2-year experiment.

**Table 2.** The concentration and removal percentage of COD during the 2-year experiment.

| Date | Concentration (mg/L) | | | Removal Percentage (%) | |
|---|---|---|---|---|---|
| | $C_i$ [1] | $C_{o,TBC}$ [2] | $C_{o,EBC}$ [3] | TBC | EBC |
| May 2019 | 55 | 38 | 28 | 31 | 49 |
| June 2019 | 66 | 44 | 45 | 33 | 32 |
| July 2019 | 63 | 42 | 35 | 33 | 44 |
| August 2019 | 48 | 36 | 29 | 25 | 40 |
| September 2019 | 35 | 21 | 25 | 40 | 29 |
| October 2019 | 47 | 28 | 22 | 40 | 53 |
| November 2019 | 42 | 21 | 24 | 50 | 43 |
| December 2019 | 37 | 19 | 10 | 49 | 73 |
| January 2020 | 33 | 20 | 14 | 39 | 58 |
| February 2020 | 25 | 12 | 8 | 52 | 68 |
| March 2020 | 25 | 15 | 9 | 40 | 64 |
| April 2020 | 36 | 17 | 12 | 53 | 67 |
| May 2020 | 42 | 18 | 11 | 57 | 74 |
| June 2020 | 54 | 21 | 10 | 61 | 81 |
| July 2020 | 63 | 24 | 8 | 62 | 87 |
| August 2020 | 40 | 15 | 9 | 63 | 78 |
| September 2020 | 36 | 18 | 10 | 50 | 72 |
| October 2020 | 38 | 14 | 8 | 63 | 79 |
| November 2020 | 49 | 21 | 11 | 57 | 78 |
| December 2020 | 47 | 18 | 12 | 62 | 74 |
| January 2021 | 50 | 17 | 9 | 66 | 82 |
| February 2021 | 61 | 20 | 10 | 67 | 84 |
| March 2021 | 47 | 16 | 11 | 66 | 77 |
| April 2021 | 39 | 12 | 13 | 69 | 67 |
| Average | 45 | 22 | 16 | 51 | 65 |
| $T$-test value | | | | 0.2 | |

[1] Inflow concentration. [2] Outflow concentration of TBC. [3] Outflow concentration of EBC.

### 3.3. Ammonium Nitrogen

The bioretention cells efficiently removed $NH_4^+$-N from inflow water. Table 3 and Figure 9 presents the test results for $NH_4^+$-N. The concentration of $NH_4^+$-N in inflow water was noticeably higher during the farming season from April to October. However, the concentration of $NH_4^+$-N in inflow water was lower from August to October 2019 because the area was affected by a drought and reduced rainfall during that time. For the removal of $NH_4^+$-N, the TBC and EBC exhibited similar performance. The average removal percentages of the TBC and EBC were 81% and 78%, respectively. Thus, both of these systems have favorable performance for $NH_4^+$-N removal. The $T$-value of $NH_4^+$-N removal between TBC and EBC was 26.0%. The test result showed that there was no difference in $NH_4^+$-N removal performance between TBC and EBC.

**Table 3.** The concentration and removal percentage of $NH_4^+$-N during the 2-year experiment.

| Date | Concentration (mg/L) | | | Removal Percentage (%) | |
|---|---|---|---|---|---|
| | $C_i$ [1] | $C_{o,TBC}$ [2] | $C_{o,EBC}$ [3] | TBC | EBC |
| May 2019 | 1.61 | 0.17 | 0.15 | 90 | 90 |
| June 2019 | 0.87 | 0.22 | 0.12 | 75 | 86 |
| July 2019 | 1.43 | 0.48 | 0.42 | 66 | 70 |
| August 2019 | 0.34 | 0.06 | 0.11 | 84 | 72 |
| September 2019 | 0.12 | 0.04 | 0.06 | 69 | 52 |
| October 2019 | 0.44 | 0.10 | 0.09 | 78 | 79 |
| November 2019 | 0.71 | 0.09 | 0.07 | 88 | 91 |
| December 2019 | 0.10 | 0.05 | 0.03 | 48 | 69 |

**Table 3.** *Cont.*

| Date | Concentration (mg/L) | | | Removal Percentage (%) | |
|---|---|---|---|---|---|
| | $C_i$ [1] | $C_{o,TBC}$ [2] | $C_{o,EBC}$ [3] | TBC | EBC |
| January 2020 | 0.12 | 0.03 | 0.03 | 77 | 75 |
| February 2020 | 0.06 | 0.02 | 0.05 | 68 | 17 |
| March 2020 | 0.27 | 0.03 | 0.08 | 90 | 71 |
| April 2020 | 0.39 | 0.07 | 0.10 | 81 | 74 |
| May 2020 | 0.71 | 0.14 | 0.17 | 81 | 76 |
| June 2020 | 0.93 | 0.18 | 0.19 | 81 | 79 |
| July 2020 | 1.63 | 0.13 | 0.12 | 92 | 92 |
| August 2020 | 1.40 | 0.19 | 0.11 | 87 | 92 |
| September 2020 | 2.49 | 0.54 | 0.35 | 78 | 86 |
| October 2020 | 1.83 | 0.27 | 0.18 | 85 | 90 |
| November 2020 | 0.62 | 0.11 | 0.16 | 82 | 75 |
| December 2020 | 0.91 | 0.08 | 0.12 | 91 | 87 |
| January 2021 | 1.08 | 0.17 | 0.14 | 84 | 87 |
| February 2021 | 0.76 | 0.10 | 0.10 | 87 | 87 |
| March 2021 | 0.84 | 0.09 | 0.09 | 89 | 89 |
| April 2021 | 1.39 | 0.10 | 0.10 | 93 | 93 |
| Average | 0.88 | 0.14 | 0.13 | 81 | 78 |
| *T*-test value | | | | 26 | |

[1] Inflow concentration. [2] Outflow concentration of TBC. [3] Outflow concentration of EBC.

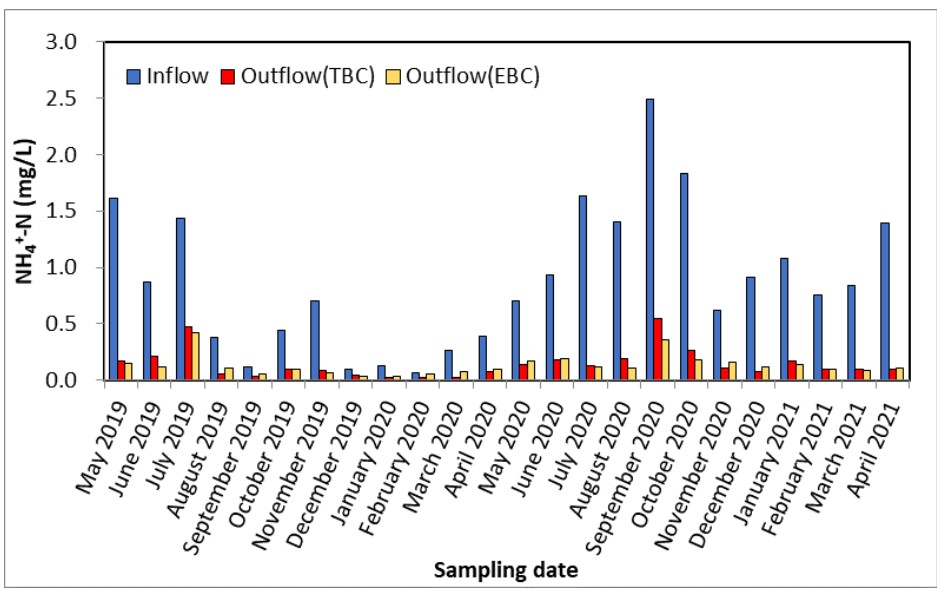

**Figure 9.** Ammonium nitrogen ($NH_4^+$-N) levels during the 2-year experiment.

### 3.4. Total Nitrogen

As indicated by Table 4 and Figure 10, under an average TN inflow of 2.23 mg/L, the average TN removal efficiency was 50% and 67% for the TBC and EBC, respectively. During the farming season, the concentration of TN was higher relative to other seasons. Nitrification coupled with denitrification is the main removal process for TN. When wastewater flows into the bioretention cells, the microorganisms in their cell systems begin to produce nitrification and denitrification reactions to reduce TN. For the TBC, a slight denitrification reaction only occurs in the planting soil layer. However, for the EBC, the denitrification reaction not only occurs in the planting soil layer but also in the mixed filter material layer. Luanmanee et al. [24] reported that the low C/N ratio of pretreated wastewater tends to result in low TN removal due to a lack of carbon available for denitrification. Consequently, in the present study, the TBC was less effective at removing TN. The addition of active charcoal powder and organic matter to the SML potentially provided a sufficient carbon

source for microorganisms. Under aerobic conditions, $NH_4^+$-N is converted into $NO_3^-$-N through biological nitrification, and the resulting $NO_3^-$-N then infiltrates into the SML, where nitrogen gas is formed through biological denitrification under anaerobic conditions and contributes to efficient TN removal (Figure 11). Therefore, the TN removal performance of the EBC is superior to that of the TBC. The *T*-value of TN removal showed that the performance of TBC and EBC in TN removal was different.

**Table 4.** The concentration and removal percentage of TN during the 2-year experiment.

| Date | Concentration (mg/L) | | | Removal Percentage (%) | |
|---|---|---|---|---|---|
| | $C_i$ [1] | $C_{o,TBC}$ [2] | $C_{o,EBC}$ [3] | TBC | EBC |
| May 2019 | 3.99 | 1.98 | 1.24 | 50 | 69 |
| June 2019 | 2.80 | 1.94 | 1.50 | 31 | 46 |
| July 2019 | 4.35 | 3.84 | 2.90 | 12 | 33 |
| August 2019 | 1.90 | 1.66 | 1.19 | 13 | 37 |
| September 2019 | 1.54 | 0.41 | 0.68 | 73 | 56 |
| October 2019 | 2.01 | 0.79 | 0.48 | 61 | 76 |
| November 2019 | 1.88 | 1.37 | 1.01 | 27 | 46 |
| December 2019 | 1.97 | 0.97 | 0.26 | 51 | 87 |
| January 2020 | 1.68 | 0.54 | 0.24 | 68 | 86 |
| February 2020 | 0.97 | 0.42 | 0.38 | 57 | 61 |
| March 2020 | 1.88 | 0.53 | 0.36 | 72 | 81 |
| April 2020 | 1.24 | 0.84 | 0.46 | 32 | 63 |
| May 2020 | 3.18 | 1.27 | 0.89 | 60 | 72 |
| June 2020 | 2.87 | 1.08 | 0.50 | 62 | 83 |
| July 2020 | 2.68 | 0.81 | 0.46 | 70 | 83 |
| August 2020 | 3.74 | 1.67 | 0.51 | 55 | 86 |
| September 2020 | 3.04 | 0.94 | 0.42 | 69 | 86 |
| October 2020 | 2.43 | 1.07 | 0.48 | 56 | 80 |
| November 2020 | 1.66 | 0.96 | 0.52 | 42 | 69 |
| December 2020 | 1.76 | 0.89 | 0.34 | 49 | 81 |
| January 2021 | 0.98 | 0.43 | 0.25 | 56 | 74 |
| February 2021 | 1.16 | 1.02 | 0.87 | 12 | 25 |
| March 2021 | 1.84 | 0.83 | 0.61 | 55 | 67 |
| April 2021 | 2.03 | 0.83 | 0.59 | 59 | 71 |
| Average | 2.23 | 1.13 | 0.71 | 50 | 67 |
| *T*-test value | | | | 0.1 | |

[1] Inflow concentration. [2] Outflow concentration of TBC. [3] Outflow concentration of EBC.

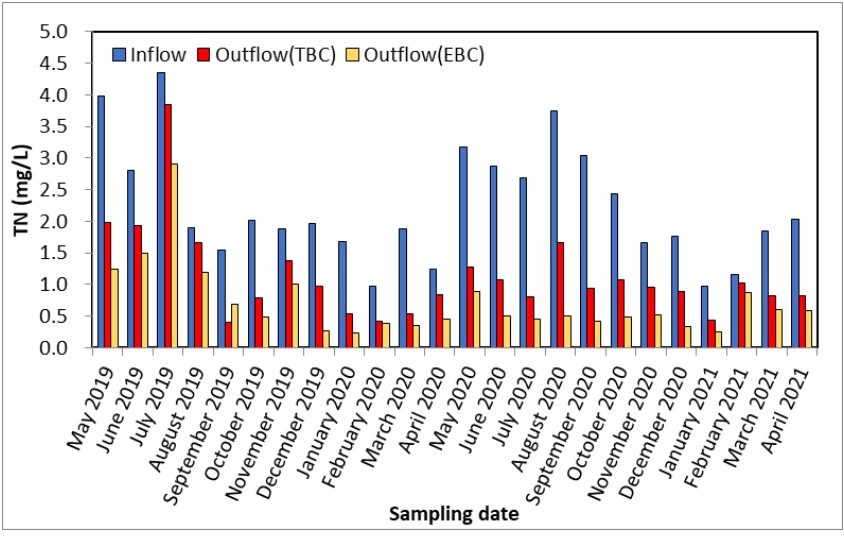

**Figure 10.** Total nitrogen (TN) levels during the 2-year experiment.

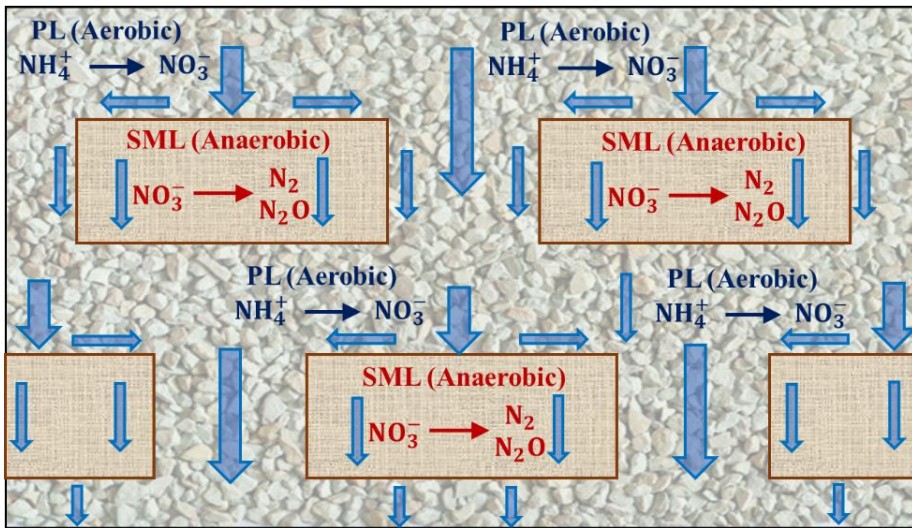

**Figure 11.** Nitrification and denitrification process in the mixed filter material layer.

To determine the nitrification and denitrification reactions in the EBC system, samples from PL and SML materials were collected (through drilling) for microbial analysis. Because the 16S rRNA gene, which is present in all bacteria, is highly conserved and can be easily amplified using universal primers, environmental microbial analyses are often performed using 16S rRNA amplicon sequencing [25]. Moreover, sequences are clustered into bins called operational taxonomic units (OTUs) on the basis of similarity. In the absence of traditional systems of biological classification, such as those available for macroscopic organisms, OTUs serve as pragmatic proxies for species (microbial or metazoan) at different taxonomic levels. For several years, OTUs have been the most commonly used units of diversity, especially for the analysis of small subunit 16S (for prokaryotes) or 18S rRNA (for eukaryotes) [26] marker gene sequence datasets. Figure 12 presents the microbial analysis results for PLs and SMLs.

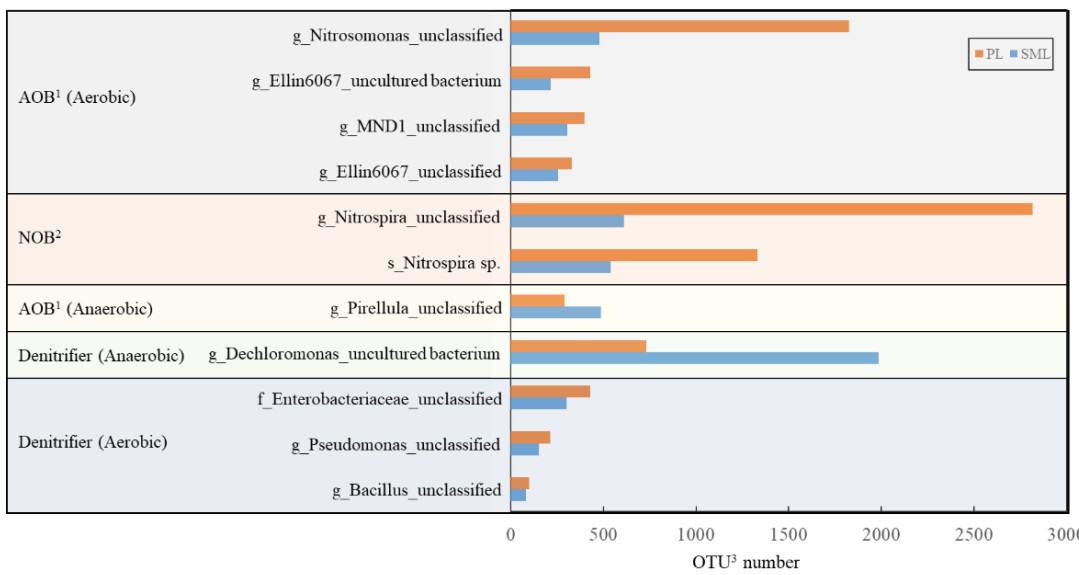

**Figure 12.** Microbial analysis results for PLs and SMLs. [1] AOB: ammonia oxidizing bacteria. [2] NOB: nitrite oxidizing bacteria. [3] OTU: Operational taxonomic unit.

The experimental results indicate that the main dominant bacteria of PLs are aerobic nitrifying bacteria. Ammonia oxidizing bacteria are responsible for oxidizing $NH_4^+$-N to $NO_2^-$-N, and nitrite oxidizing bacteria (NOB) are responsible for oxidizing $NO_2^-$-N to $NO_3^-$-N. In PLs, the nitration reaction can be completed through the two aforementioned

reactions. Therefore, both the TBC and EBC provide a favorable nitration reaction for $NH_4^+$-N. However, the denitrification reaction is mainly influenced by anaerobic denitrifying bacteria. The test results indicate that the anaerobic bacteria in SMLs consisted of 2818 OTUs, which is 4.6 times the amount detected in the PLs (612 OTUs). When the agricultural runoff water with $NH_4^+$-N flows into the mixed filter material layer, the aerobic bacteria of the PLs first nitrify the $NH_4^+$-N into $NO_2^-$-N and $NO_3^-$-N. Subsequently, the water flows into the SMLs (anaerobic zone), where it undergoes denitrification, which reduces $NO_3^-$-N and $NO_2^-$-N to $N_2$. Therefore, the TN removal performance of the EBC is superior to that of the TBC.

### 3.5. Total Phosphorus and Phosphate

The phosphorus in the runoff from agricultural land is a key component of nonpoint source pollution, and it can accelerate the eutrophication of lakes and streams [27]. Therefore, the effective reduction of the phosphorus in agricultural runoff is a crucial task. Table 5 and Figure 13 chart the TP concentrations of the inflow into the cells, the outflow from the TBC, and the outflow from the EBC. The nonpoint source pollution during the fertilization period contained a relatively high TP concentration relative to the subsequent period. The average concentration of TP in inflowing water was 0.761 mg/L. After TBC purification, the average concentration of TP in the outflow was 0.362 mg/L, indicating a removal percentage of 51%. However, the average concentration of TP after EBC purification was 0.059 mg/L, indicating a removal percentage of 92%. These results indicate that the TP removal effect of the EBC is superior to that of the TBC. The *T*-value of TP removal between TBC and EBC was <0.1%. It showed that the performance of TBC and EBC in TP removal was significantly different.

**Table 5.** The concentration and removal percentage of TP during the 2-year experiment.

| Date | Concentration (mg/L) | | | Removal Percentage (%) | |
|---|---|---|---|---|---|
| | $C_i$ [1] | $C_{o,TBC}$ [2] | $C_{o,EBC}$ [3] | TBC | EBC |
| May 2019 | 0.56 | 0.27 | 0.05 | 52 | 91 |
| June 2019 | 0.81 | 0.53 | 0.05 | 35 | 94 |
| July 2019 | 0.67 | 0.32 | 0.07 | 52 | 90 |
| August 2019 | 0.91 | 0.26 | 0.07 | 71 | 92 |
| September 2019 | 0.80 | 0.37 | 0.05 | 54 | 94 |
| October 2019 | 0.94 | 0.27 | 0.07 | 71 | 93 |
| November 2019 | 0.72 | 0.42 | 0.09 | 42 | 88 |
| December 2019 | 0.68 | 0.33 | 0.05 | 51 | 93 |
| January 2020 | 0.51 | 0.29 | 0.04 | 43 | 92 |
| February 2020 | 0.84 | 0.40 | 0.02 | 52 | 98 |
| March 2020 | 0.89 | 0.32 | 0.02 | 64 | 98 |
| April 2020 | 0.70 | 0.41 | 0.04 | 41 | 98 |
| May 2020 | 0.99 | 0.48 | 0.05 | 52 | 95 |
| June 2020 | 1.03 | 0.52 | 0.07 | 50 | 93 |
| July 2020 | 0.63 | 0.31 | 0.06 | 51 | 90 |
| August 2020 | 0.74 | 0.37 | 0.10 | 50 | 86 |
| September 2020 | 0.94 | 0.42 | 0.05 | 55 | 95 |
| October 2020 | 0.83 | 0.31 | 0.08 | 63 | 90 |
| November 2020 | 0.56 | 0.33 | 0.05 | 41 | 91 |
| December 2020 | 0.67 | 0.29 | 0.05 | 57 | 93 |
| January 2021 | 0.58 | 0.34 | 0.05 | 41 | 91 |
| February 2021 | 0.60 | 0.41 | 0.07 | 32 | 88 |
| March 2021 | 0.74 | 0.33 | 0.10 | 55 | 86 |
| April 2021 | 0.93 | 0.38 | 0.09 | 59 | 90 |
| Average | 0.76 | 0.36 | 0.06 | 51 | 92 |
| *T*-test value | | | | <0.1 | |

[1] Inflow concentration. [2] Outflow concentration of TBC. [3] Outflow concentration of EBC.

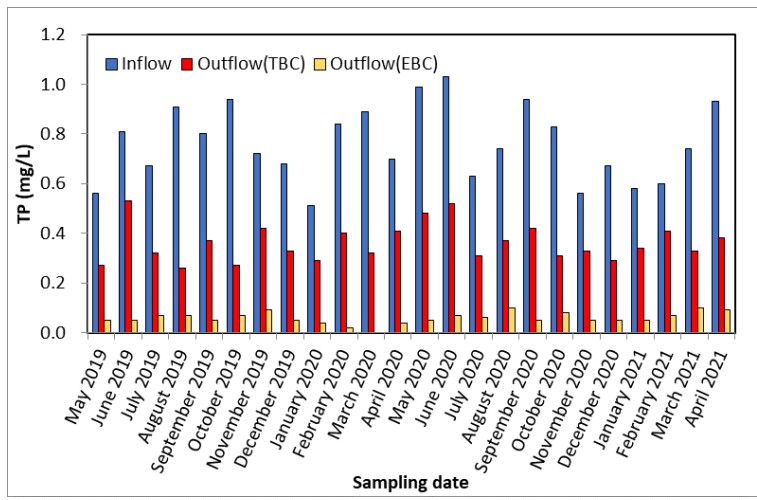

**Figure 13.** Total phosphorus (TP) levels during the 2-year experiment.

Table 6 and Figure 14 presents the performance of the TBC and EBC for the removal of phosphate ($PO_4^{3-}$) in water. The average concentration of phosphate in inflow was 0.625 mg/L. However, the phosphate removal percentage of the TBC (49%) was significantly higher than that of the EBC (90%), indicating that the EBC was more effective than the TBC in removing phosphate. In the TBC, the phosphate removal percentage differed significantly over time. In the EBC, the phosphate removal percentage did not exhibit significant changes over time. This indicates that the EBC is more stable than the TBC in removing phosphate. The *T*-value of $PO_4^{3-}$ removal also showed that the performance of TBC and EBC in $PO_4^{3-}$ removal was significantly different.

**Table 6.** The concentration and removal percentage of $PO_4^{3-}$ during the 2-year experiment.

| Date | Concentration (mg/L) | | | Removal Percentage (%) | |
|---|---|---|---|---|---|
| | $C_i$ [1] | $C_{o,TBC}$ [2] | $C_{o,EBC}$ [3] | TBC | EBC |
| May 2019 | 0.67 | 0.38 | 0.05 | 43 | 93 |
| June 2019 | 0.67 | 0.34 | 0.05 | 49 | 93 |
| July 2019 | 0.72 | 0.41 | 0.07 | 43 | 90 |
| August 2019 | 0.72 | 0.43 | 0.07 | 39 | 90 |
| September 2019 | 0.75 | 0.45 | 0.05 | 41 | 93 |
| October 2019 | 0.71 | 0.40 | 0.07 | 43 | 90 |
| November 2019 | 0.62 | 0.31 | 0.05 | 50 | 92 |
| December 2019 | 0.55 | 0.25 | 0.04 | 55 | 93 |
| January 2020 | 0.40 | 0.21 | 0.06 | 48 | 85 |
| February 2020 | 0.74 | 0.42 | 0.03 | 43 | 96 |
| March 2020 | 0.67 | 0.28 | 0.02 | 58 | 97 |
| April 2020 | 0.59 | 0.31 | 0.04 | 47 | 93 |
| May 2020 | 0.72 | 0.37 | 0.06 | 49 | 92 |
| June 2020 | 0.75 | 0.42 | 0.07 | 44 | 91 |
| July 2020 | 0.54 | 0.27 | 0.05 | 50 | 91 |
| August 2020 | 0.59 | 0.31 | 0.08 | 47 | 86 |
| September 2020 | 0.74 | 0.31 | 0.05 | 58 | 93 |
| October 2020 | 0.71 | 0.26 | 0.08 | 63 | 89 |
| November 2020 | 0.41 | 0.23 | 0.05 | 44 | 88 |
| December 2020 | 0.51 | 0.22 | 0.06 | 57 | 88 |
| January 2021 | 0.41 | 0.31 | 0.05 | 24 | 87 |
| February 2021 | 0.44 | 0.24 | 0.07 | 45 | 84 |
| March 2021 | 0.59 | 0.22 | 0.08 | 63 | 86 |
| April 2021 | 0.79 | 0.28 | 0.09 | 65 | 89 |
| Average | 0.63 | 0.32 | 0.06 | 49 | 90 |
| *T*-test value | | | | <0.1 | |

[1] Inflow concentration. [2] Outflow concentration of TBC. [3] Outflow concentration of EBC.

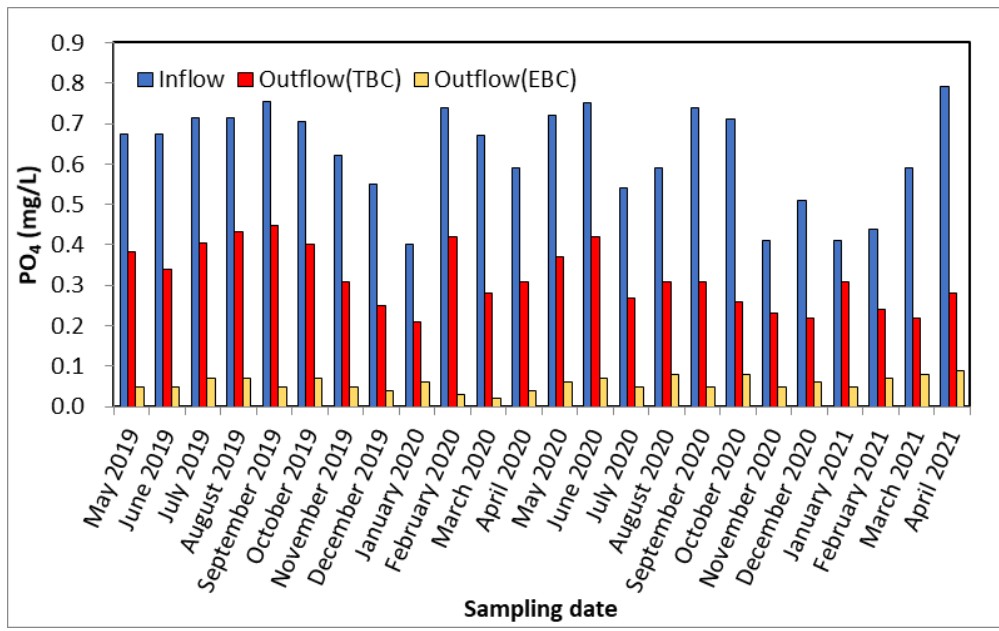

**Figure 14.** Phosphate ($PO_4^{3-}$) levels during the 2-year experiment.

Normally, the phosphorus in agricultural runoff contains particulate and dissolved forms. The particulate phosphorus in water can be removed through filtering. Therefore, both the TBC and EBC can remove particulate phosphorus through the filtration of water in the upper planting soil. However, dissolved phosphorus is generally difficult to remove through filtration. In the EBC, a specific ratio of iron particles is mixed into its SMLs. Phosphorus can be adsorbed by the Al and Fe hydroxides in the soil. As discussed in an earlier section of this paper, the inside of SMLs presents an anaerobic state (Figure 15). In such an environment, the iron added to SMLs transforms into ferrous iron ($Fe^{2+}$), which is subsequently translocated to PLs and oxidized to ferric ion ($Fe^{3+}$); $Fe^{3+}$ aids the coprecipitation of $PO_4^{3-}$ from percolating wastewater [28]. Under anaerobic conditions, the reaction mechanism through which $PO_4^{3-}$ produces $Fe_3(PO_4)_2$ is expressed by Equation (1). Equations (2) and (3) express the aforementioned reaction mechanism of $PO_4^{3-}$ under aerobic conditions. The final reaction product of iron ions and phosphate precipitates at the bottom of a bioretention cell. Sato et al. [29] monitored a case in Japan using SMB to treat domestic sewage and used mass balance to calculate TP removal efficiency. It shows that 110 kg of metal iron fix 52 kg of P, and its service life is more than 10 years. The SML of this study used the same percentage of metal iron content as in the previous study. As the TP concentration of agriculture non-point source pollution is much lower than that of domestic sewage, the service life of this study site can be expected to be longer than 10 years.

$$3Fe^{2+} + 2PO_4^{3-} \rightarrow Fe_3(PO_4)_2 \ (\downarrow) \tag{1}$$

$$Fe^{3+} + PO_4^{3-} \rightarrow FePO_4 \ (\downarrow) \tag{2}$$

$$Fe(OH)_3 + H_3PO_4 \rightarrow FePO_4 \ (\downarrow) + 3H_2O \tag{3}$$

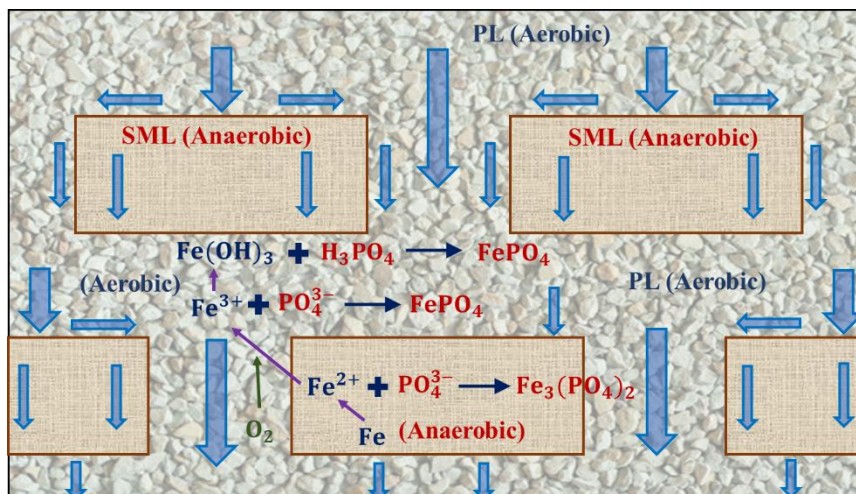

**Figure 15.** Iron ions and phosphate reaction process in the mixed filter material layer.

## 4. Conclusions

The results of the present study indicate that both the TBC and EBC can purify agricultural nonpoint source pollution. The removal percentages for SS, COD, and $NH_4^+$-N were similar in the TBC and EBC. However, for TN, TP, and $PO_4^{3-}$ removal, the EBC significantly outperforms the TBC. Regarding the composition of the two bioretention cells, the only difference is the bottom material. The TBC uses a gravel bed, whereas the EBC uses a mixed filter material layer. The two layers that form the mixed filter material layer comprise an aerobic layer and an anaerobic layer. The aerobic layers (PLs) consist of zeolite, and they alternate with the anaerobic layers (SMLs). The zeolite in the aerobic layers increases the adsorption of $NH_4^+$-N and reacts to nitrate through the reaction of nitrifying bacteria. Moreover, zeolite can enhance the oxidation and precipitation of mobile ferrous iron to high-surface-area ferric oxide, which improves phosphorus sorption. In the anaerobic layer formed by SMLs, nitrate is converted into nitrous oxide and nitrogen (denitrification), and ferrous iron is oxidized to mobile ferric iron, which exits the anaerobic layer and reacts to $FePO_4$ with the $PO_4^{3-}$ in the aerobic layer. As a result, relative to the TBC, the EBC provides superior removal performance for TN, TP, and $PO_4^{3-}$. The use of the EBC to purify agricultural nonpoint source pollution results in greater pollution removal efficiency, and consequently, contributes to SDGs such as environmental sustainability, clean water, and food security.

**Author Contributions:** C.-C.H.: Conceptualization, writing manuscript draft and supervision; Y.-X.L.: Investigation, formal analysis and data curation. All authors have read and agreed to the published version of the manuscript.

**Funding:** This research received no external funding.

**Informed Consent Statement:** Informed consent was obtained from all subjects involved in the study.

**Data Availability Statement:** Data used in this study are duly available from the first authors on reasonable request.

**Acknowledgments:** The authors are grateful to the Editor and anonymous reviewers for their constructive comments and suggestions.

**Conflicts of Interest:** The authors declare no conflict of interest.

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
