# Peer review of "Pollutant Removal Efficiency of a Bioretention Cell with Enhanced Dephosphorization"

_water, doi:10.3390/w14030396_

Round 1

Reviewer 1 Report

Thank you for allowing me to review this article.

The manuscript entitled "Pollutant Removal Efficiency of a Bioretention Cell with Enhanced Dephosphorization" is an original contribution, and the topic is interesting for readers of the Water journal.

The presentation is fine but I noticed criticism in the text that should be addressed accordingly. Authors can use the comments below to improve their article.

1) I suggest changing the title of figure 2 to "Figure 2. Profile of the retention cell (a) traditional, b) inverted).

2) I suggest adding in the text the geographic coordinates of the study area.

Author Response

The manuscript entitled "Pollutant Removal Efficiency of a Bioretention Cell with Enhanced Dephosphorization" is an original contribution, and the topic is interesting for readers of the Water journal.

The presentation is fine but I noticed criticism in the text that should be addressed accordingly. Authors can use the comments below to improve their articles.

Point 1: I suggest changing the title of figure 2 to "Figure 2. Profile of the retention cell (a) traditional, b) inverted).

 Response 1: We thank the reviewer for the very interesting comment. The precedent version of the title of figure 2 has been replaced, becoming “Figure 2. Profile of the retention cell (a) traditional, (b) inverted. (Line 76, page 3)

Point 2: I suggest adding in the text the geographic coordinates of the study area.

Response 2: Thank you for the suggestion. We have added the geographic coordinates as explained above (Lines 120-121, page 4). “GPS coordinates of the study site is N31°40.01’, E117°40.73’.”

Reviewer 2 Report

See the enclosed file.

Author Response

The study presents a comparison of a bioretention cell with enhanced dephosphorization (EBC) with a traditional bioretention cell (TBC) for the removal of nonpoint source pollution. The lowermost layer of the EBC was a mixed filter material (native soil + active charcoal powder, organic matter, and iron) layer instead of a gravel bed, as is the case in TBCs. The cells of dimensions 45 x 15 x 1.2 m were investigated for 2 years in the field. Sampling was performed once a month, and a total of 24 x 3 = 72 samples were collected for analysis. Suspended solids (SS), chemical oxygen demand (COD), ammonium nitrogen (N-NH4+), total nitrogen (TN), total phosphorus (TP), and phosphate (PO-4) were tested in the inflow (Ci) and both outflows (Co EBC and Co TBC).

Point 1: The experiment is valuable due to its large scale and a relatively long time of monitoring. However, it is known that typically, the phosphorus removal efficiency in bioretention cells is decreasing in time. Trends in removal efficiency (? =100 (Ci-Co)/Ci) would be welcome to show how stable has been the efficiency in time.

Response 1: Thank you for the suggestion. We have added the information required as explained above. “Sato et al. monitored a case in Japan using SMB to treat domestic sewage and used mass balance to calculate TP removal efficiency. It shows that 110 kg of metal iron fix 52 kg of P, and its service life is more than 10 years. The SML of this study used the same percentage of metal iron content as in the previous study. Since the TP concentration of agriculture non-point source pollution is much lower than that of domestic sewage, the service life of this study site can be expected to be longer than 10 years.” (Line 337-342, page 16)

Point 2: My second remark concerns the statistical analysis. The authors gathered good material to compare EBC with TBC using a statistical Student’s t-test. The null hypothesis is that there is no difference between the means ?? ???= ?? ???. This test allows for finding a significant difference when the null hypothesis is false. I strongly recommend making 6 tests (for all analyzed pollutants) to get more justified conclusions.

Response 2: Thank you for underlying this deficiency. We have added the Student’s t-test value for all analyzed pollutants (Table 1~Table 6).

Point 3: The last but not least, the removal percentage rounding should be to the nearest integers, not to the decimal digit, e.g.: 51% instead of 51.4%.

Response 2: We have modified the removal percentage to the nearest integers according to the comment.

Author Response

The issue of the study is to improve the capability of a LID, bioretention cell, and to provide its’ efficiency from field measurement. It is not difficult to find papers regarding water quality and quantity estimations in watersheds, however, it is not easy to find how much flow and NPS loads could be reduced with BMP or LID practice applications. Moreover, many hydrological models provide an opportunity to simulate the practices, still, there is a need to test if BMP/LID efficiencies are correct or applicable. Thus, the study is considered one of the good cases providing scientific evidence.

Thus, the reviewer enjoyed reading the manuscript, the manuscript is almost ready to be published. But still, there is a need to revise or improve some minor things. Please see the specific comments.

 Point 1: Figure 3: The definitions of SML and PL are described in the text though, it would be better to make this clear in the figure, maybe adding a zoom-in figure.

 Response 1: Thank you for underlying this deficiency. Figure 3 was modified according to the comment. (Line 97, page 4)

Point 2: Result section: If it is possible, how about providing actual numbers of the measurements with tables? The authors already provided the results with figures 7-10 and 13-14, and removal percentages are provided. If tables are given with the figures, it will be very helpful for other researchers.

Response 2: Thank you for the suggestion. We have added Table 1 ~ Table 6 to show the actual data of the measurements.